# Learning Lightweight Structure-Aware Embeddings for Protein Sequences

**Sidharth Lakshmanan, Jeff Nivala**
Paul G. Allen School of Computer Science and Engineering
University of Washington
Seattle, WA, USA
{sidlak,jmdn}@cs.washington.edu

**Philip J. Y. Leung**
Department of Biochemistry
University of Washington
Seattle, WA, USA
pleung@uw.edu

## Abstract

Machine learning models, such as AlphaFold, have recently demonstrated remarkable accuracy in predicting the structures of protein sequences. This capability enables their use as oracles for providing structure-based information to aid other learning tasks. In this study, we investigate the use of deep learning embeddings and explore the feasibility of developing a structure-aware protein sequence embedding. To accomplish this, we employ S4PRED and ESMFold, two models that predict protein secondary (2D) and tertiary structures (3D) respectively, directly from single sequences. These models act as oracles to form structure-aware embeddings through an autoencoder. We then compare this approach to purely sequence-based embeddings in a Protein-Protein Interaction (PPI)-prediction task. Our findings highlight the potential advantages of employing structure embeddings and provide grounds for future research directions.

## 1 Introduction

Deep learning models are extensively used to predict and generate information about proteins. To do this, these models use an embedding, which converts the protein sequence into a numerical form that can be used for training. The most classical of these embeddings are: one-hot encoding, where each of the 20 amino acids are encoded as a 20-dimensional vector with a single bit set to 1; BLOSUM62 encoding, based on the 20-dimensional BLOSUM Matrix which is derived from clustering of aligned segments in blocks (Steven Henikoff, 1992); and VHSE8 encoding, an 8-dimensional matrix that contains hydrophobic, steric, and electronic properties (Hu Mei, 2005). While all of these embedding matrices are useful, none of them provide any higher-level structural information about a protein. This means that for deep learning tasks in which protein structure is important, for example in predicting protein-protein interactions (PPIs) the network needs to first learn about the structure of the protein before starting to become more specialized. In this work, we begin exploring this issue by using an autoencoder to generate a structure embedding for protein sequences. We do so by creating two separate embeddings: one using a secondary structure-prediction model called S4PRED (Lewis Moffat, 2021) and another using a tertiary structure-prediction model called ESMFold (Lin et al., 2022). We show that this concept is feasible and worth pursuing by comparing our structure embedding to classical embeddings on the deep learning task of PPI prediction, a phenomenon that occurs through structural interactions between two proteins (ElAbd H., 2020).

## 2 Methods

We trained three types of embeddings: one structure embedding using S4PRED, one structure embedding using ESMFold, and one pure sequence embedding; see Appendix section 5.3 for full model structure and training details. The dataset used to train the autoencoder consisted of 97,616 proteins of length 600 amino acids or smaller from the Protein Database or PDB (Berman et al., 2000); see Appendix section 5.4 for full dataset details.

Table 1: Comparison of Encodings on PPI Prediction

| Encoding | Train Acc | Train AUC | Test Acc | Test AUC | Flattened Size |
|---|---|---|---|---|---|
| One Hot (20 dimensions) | 0.86 | 0.93 | 0.83 | 0.88 | 12000 |
| BLOSUM (20 dimensions) | 0.99 | 0.99 | 0.80 | 0.86 | 12000 |
| VHSE8 (8 dimensions) | 0.81 | 0.89 | 0.80 | 0.86 | 4800 |
| Structure Embedding w/ S4PRED | 0.71 | 0.75 | 0.69 | 0.80 | 100 |
| Structure Embedding w/ ESMFold | 0.82 | 0.84 | 0.80 | 0.76 | 100 |
| Pure Sequence | 0.83 | 0.83 | 0.81 | 0.83 | 100 |

## 2.1 STRUCTURE EMBEDDING WITH S4PRED

The loss of the S4PRED-based autoencoder uses a comparison of the input sequence's secondary structure as computed by DSSP (Kabsch & Sander, 1983) and S4PRED's prediction of the secondary structure on the output protein sequence. Both secondary structures were represented as one-hot encoded sequences of $[C|E|H] * n$ and were compared using a categorical cross entropy loss. For back-propagation, we ensured that the weights of S4PRED were frozen during training. Refer to the diagram in the Appendix 5.1 for a visualization of this model structure.

## 2.2 STRUCTURE EMBEDDING WITH ESMFOLD

The loss of the ESMFold-based autoencoder uses Pyrosetta's (Chaudhury et al., 2010) implementation of TM-Score to align and evaluate the similarity between the atomic coordinates of the experimental structure and the atomic coordinates of the ESMFold predicted structure. Again, we ensured that the weights of ESMFold were frozen during back-propagation. Refer to the diagram in the Appendix 5.2 for a visualization of this model structure.

## 2.3 PURE SEQUENCE EMBEDDING

Instead of using an oracle, the cross entropy loss was simply calculated based on the one-hot encodings of the input and output sequences directly.

## 3 RESULTS

Table 1 shows results of the PPI prediction task for all of the encodings that were benchmarked. Acc is the accuracy of the PPI model and AUC is the area under the ROC curve of the model. The flattened size of each encoding is the number of values needed to represent the protein sequence. Refer to the Appendix section 5.5 for more details on how these encodings were benchmarked. Overall, the data indicates that this approach of creating a lightweight structure-aware embedding, especially in ESMFold case, was able to provide accuracy comparable to classical embeddings and a pure sequence embedding while being much more memory efficient relative to the former.

## 4 DISCUSSION

By incorporating oracles like S4PRED or ESMFold, we are able to train the encoding to incorporate both sequence and structural features, giving advantages in terms of computational performance without sacrificing too much accuracy. The accuracy and memory usage achieved by the lightweight structure-aware encoding is promising, especially considering that the model utilized only the secondary/tertiary structures of the proteins for PPI prediction. This underscores the potential for improving the accuracy of structure-aware encoding through a hybrid approach, optimizing for simultaneous recovery of sequence and structural information. Alternatively, task-specific structure-aware encodings could be generated to optimize the encoding for specific tasks. In conclusion, this work on structure-aware encoding serves as a foundation for future development in this area, with promising avenues for enhancing the accuracy of the encoding through a hybrid approach and task-specific design.

URM STATEMENT

The authors acknowledge that at least one key author of this work meets the URM criteria of ICLR 2023 Tiny Papers Track.

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

## 5 APPENDIX

The following sections contain more details on how the encodings were trained and benchmarked.

## 5.1 S4PRED-BASED AUTOENCODER

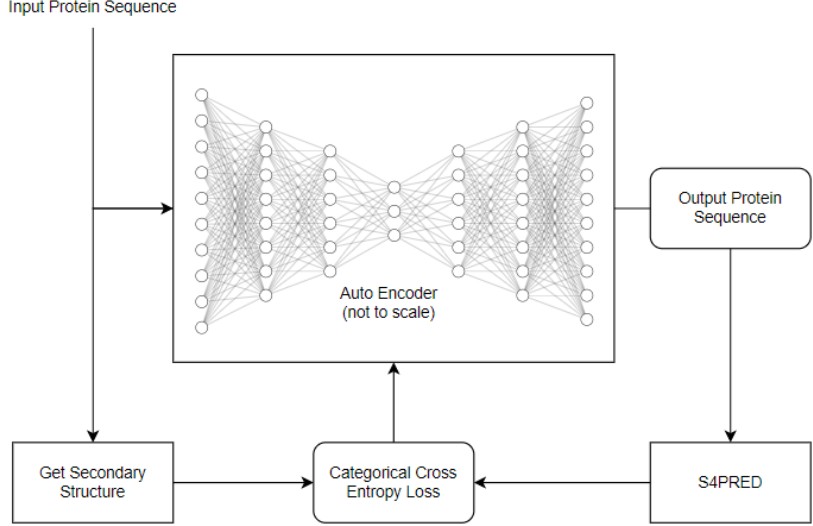

## 5.2 ESMFOLD-BASED AUTOENCODER

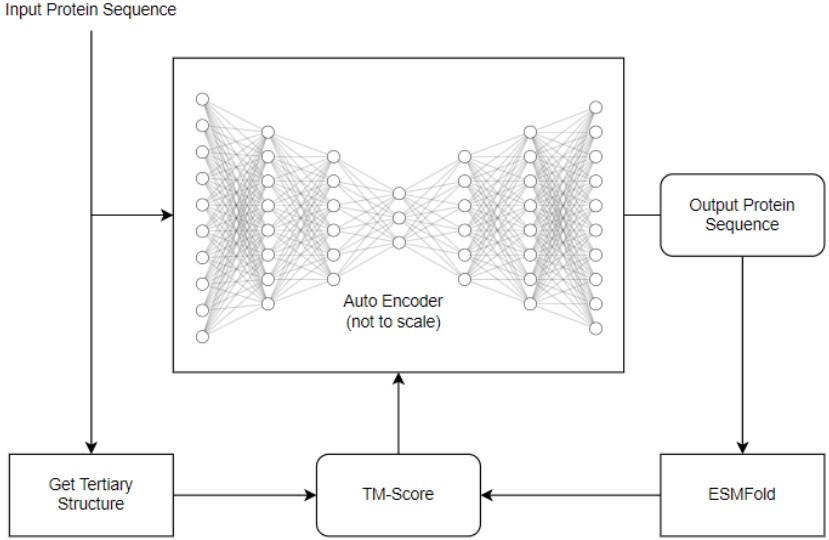

## 5.3 MODEL TRAINING AND STRUCTURE

The embeddings were trained using an autoencoder model in the Pytorch ML framework. The encoder part of the model is 6 fully connected linear hidden layers with LeakyReLU's between each of its layers with layer widths of 10000, 5000, 1000, 500, 300, 100. The embedding size of the encoder was set to 100. The decoder part of the model is 5 fully connected linear hidden layers with LeakyReLU's between each of the layers and a Sigmoid activation function on the final layer mirroring the widths of the encoder. Stochastic gradient descent was used as the Optimizer. Training of the autoencoder was carried out using a bootstrapping inspired technique using batches of 100000 sequences drawn uniformly at random. We trained this model using 6 RTX6000 GPU's.

## 5.4 DATASET

The dataset used to train the autoencoder was sequences from the Protein Database (PDB). We chose this dataset because it contained proteins with defined primary and secondary structures, which is not a guarantee for proteins in sequence-only databases such as Uniref. We then filtered out any proteins with sequences of length 600 or higher and padded every sequence with dashes to a length of 600. Finally, we filtered out all of the sequences with unknown regions (determined by an $X$ in the sequence or structure). As a result, we ended up with 97,616 proteins in the dataset.

## 5.5 BENCHMARKING

Using Pytorch, we re-implemented an architecture inspired by (Somaye Hashemifar, 2018) for identifying PPIs similar to the model used by (ElAbd H., 2020). This model is composed of two main parts. The first part is a convolution neural network consisting of 4 modules with a convolutional layer, followed by a rectified linear unit (ReLU), a batch normalization layer, and an average pooling layer in all except the final layer which using a pooling layer for the Global average. Every forward pass takes a pair of proteins and generates a pair of vectors that represent each of these vectors using the same weights. The second part of this mode is a feed-forward multilayer perceptron which receives the pair of vectors and outputs the probability that the pair of proteins interact, thereby predicting PPI. Like (ElAbd H., 2020), we also make the optimization of using 4 modules instead of 5 to speed up training time and allow for faster experimentation. Note that this model uses an Adam optimizer and a sigmoid function for the activation function for each of the prediction layers. Finally, binary cross-entropy was used as a loss function.

## 5.6 CODE

For source code, visit

https://github.com/uwmisl/Deep-Molecules

