# OpenReview forum: "LEARNING LIGHTWEIGHT STRUCTURE-AWARE EMBEDDINGS FOR PROTEIN SEQUENCES"
_ICLR.cc/2023/TinyPapers — Submitted to Tiny Papers @ ICLR 2023_

### Official Review · Reviewer_Asxq · 2023-03-28

**Confidence:** 4

**Summary Of Contributions:**

This paper tackles the question of improving protein sequence embeddings with pre-trained models that include knowledge of secondary structure. The method trains an autoencoder whose loss is based on MSE in a secondary structure encoded space using S4PRED.

**Rating:**

Great Start (GS): a submission which meets some of the reviewing criteria but has room for improvement

**Strengths And Weaknesses:**

Strengths:

- An interesting idea to improve protein embeddings using a pre-trained secondary structure model.
- The method is clear and (mostly reproducible)
- This paper meets the form requirements

Weaknesses:

- Would be nice to include widths of the 6 layers (as well as depth)
- The structure aware embeddings don’t seem to help on the PPI prediction task.

**Suggested Changes:**

- I’m not certain on the output of S4PRED, I assume its a graph? It would be great to clarify this.
- From reading the paper I thought the current results showed the structure aware model working better than a sequence only model. This seems to be not the case. It would be great to make the claims match the experiments. I still think this is an interesting direction, though other architectures or ways to incorporate structure information may be more fruitful.

Some suggestions to improve embedding

- I would encourage the authors to try other ways of encoding information contained in S4PRED into the embeddings. Such as using the output of an intermediate layer as information for the embedding. I think the current method is difficult as it loses any information unnecessary for secondary structure prediction. Thus if there is any sequence level information not included in the (predicted) secondary structure that is useful for PPI prediction, then this will be lost in the “Structure-Aware” embedding. I would call the current model a “Structure-Only” model and would be interested in a combination with the pure sequence model.
- The current autoencoder is very large for a dense model, I believe there are some training challenges with this structure. You may want to try convolutional structure or some more parameter efficient model.

---

> ### Author Response · Authors · 2023-05-31
> **We have addressed the reviewer's comments by training smaller models and introducing ESMFold as another oracle.**
>
> Thank you for taking your time to review our paper. We greatly appreciate your feedback and have revised our paper to address all of the weaknesses and comments.
>
> * Q. Would be nice to include widths of the 6 layers (as well as depth)
>
> A. We have now added this information to Appendix Section 5.3 titled "Model Training and Structure"
>
> * Q. I’m not certain on the output of S4PRED, I assume its a graph? It would be great to clarify this.
>
> A. We have now clarified this. The output of S4PRED is a sequence of C's, E's, and H's corresponding to the secondary structure of the protein sequence.
>
> * Q. From reading the paper I thought the current results showed the structure aware model working better than a sequence only model. This seems to be not the case. It would be great to make the claims match the experiments. I still think this is an interesting direction, though other architectures or ways to incorporate structure information may be more fruitful.
>
> A. We thank the reviewer for validating our interest in this approach. We now show the ESMFold-based structure aware method as nearly competitive with one-hot encoding and comparable to the BLOSUM and VHE8 methods, despite substantially reduced dimensionality.
>
> * Q. I would encourage the authors to try other ways of encoding information contained in S4PRED into the embeddings. Such as using the output of an intermediate layer as information for the embedding. I think the current method is difficult as it loses any information unnecessary for secondary structure prediction. Thus if there is any sequence level information not included in the (predicted) secondary structure that is useful for PPI prediction, then this will be lost in the “Structure-Aware” embedding. I would call the current model a “Structure-Only” model and would be interested in a combination with the pure sequence model.
>
> A. We agree with the reviewer that an intermediate layer of the S4PRED model could provide a useful representation of both sequence and structure. However, our main goal in this work was to explore and benchmark the extent to which structure by itself (structure-only, as the reviewer puts it) could perform as an embedding. Which, using the 3D structural embedding as obtained from ESMfold, does surprisingly well. We also include a Sequence-Only embedding to the new results, and point towards future work that look at embeddings which combine both sequence and structural information.
>
> * Q. The current autoencoder is very large for a dense model, I believe there are some training challenges with this structure. You may want to try convolutional structure or some more parameter efficient model.
>
> A. Our revisions are now using models that have fewer layers and reduced width, which indeed made training faster and more efficient.

---

### Official Review · Reviewer_3PT5 · 2023-03-30

**Confidence:** 2

**Summary Of Contributions:**

The paper under review explores the feasibility of developing a structure-aware embedding for protein sequences. The authors used an autoencoder to generate the embeddings and employed a relatively simple secondary structure prediction model, S4PRED, as an oracle. The authors trained two types of embeddings - structure-aware and purely sequence aware - and compared their performance in a protein-protein interaction prediction task.

**Rating:**

Needs Clarification (NC): a submission which does not meet the reviewing criteria and needs clarification for its described problem or solution

**Strengths And Weaknesses:**

## Strengths

- The paper addresses an important problem in the field of deep learning for protein research - the incorporation of high-level structural information in protein representation learning.

## Weaknesses

-  The paper does not compare the performance of the proposed structure-aware embedding to state-of-the-art protein structure prediction models such as AlphaFold, which limits the scope of the findings. Since the proposed approach performs much worse than the baselines, I do not see why the authors do not use a more sophisticated structure learning model.



**Suggested Changes:**

I would suggest the authors to investigate more sophisticated autoencoders or use the publicly available AlphaFold model. Moreover, some parts of the text are not written clearly (for example there is no definition of  S4PRED). The authors may need to clarify the definition of S4PRED or any other technical terms they use in their paper. This will help readers better understand the methods they used and the results they obtained. Additionally, if they are using a modified version of S4PRED, they should provide details on the modifications they made and the reasons behind them.

---

> ### Author Response · Authors · 2023-05-31
> **We have addressed the reviewer's comments by rewriting our paper to be more clear and introducing ESMFold as a way to get 3D structure prediction.**
>
> Thank you for taking your time to review our paper. We greatly appreciate your feedback and have revised our paper to address all of the weaknesses and comments.
>
> * Q. I would suggest the authors to investigate more sophisticated autoencoders or use the publicly available AlphaFold model.
>
> A. We thank the reviewers for this discussion and have incorporated ESMfold as an alternative approach to 3D structure prediction. ESMfold is designed and optimized for the use case of single sequence predictions, as we are doing here. AlphaFold is intended to be used with multiple sequence alignments, which was not feasible for our approach.
>
> * Q. Moreover, some parts of the text are not written clearly (for example there is no definition of S4PRED). The authors may need to clarify the definition of S4PRED or any other technical terms they use in their paper. This will help readers better understand the methods they used and the results they obtained.
>
> A. We apologize for the lack of clarity and have rewritten the paper to be more clear.
>
> * Q. Additionally, if they are using a modified version of S4PRED, they should provide details on the modifications they made and the reasons behind them.
>
> A. We did not modify the code for S4PRED or ESMFold, and did not modify the weights for these oracles during training. We now also share code used for the model at https://github.com/uwmisl/Deep-Molecules/

---

### Official Review · Reviewer_uMJN · 2023-03-31

**Confidence:** 4

**Summary Of Contributions:**

This paper proposes a structure-aware protein sequence embedding using an autoencoder. The authors use a neural network model (S4PRED) to generate structure-aware embeddings and compare this approach to purely sequence-based embeddings in a Protein-Protein Interaction (PPI)-prediction task.

**Rating:**

Clear, Correct, and Reproducible (CCR): a submission which meets the reviewing criteria

**Strengths And Weaknesses:**

Strength:

- The authors show that structure-aware embeddings have potential advantages and provide grounds for future research directions.

Weaknesses:

- While the logic of the paper seems interesting, the proposed method failed to outperform or perform on-par with classical and existing methods.
- The authors should be able to explain why they think the structure-aware method did not perform better than the classical methods as expected. This information will be really helpful for other works.


**Suggested Changes:**

See weaknesses

---

> ### Author Response · Authors · 2023-05-31
> **We have addressed the reviewer's concerns about the performance of our encodings by introducing an ESMFold-based autoencoder.**
>
> Thank you for taking your time to review our paper. We greatly appreciate your feedback and have revised our paper to address all of the weaknesses.
>
> * Q. While the logic of the paper seems interesting, the proposed method failed to outperform or perform on-par with classical and existing methods.
>
> A. We acknowledge that the S4pred-based method underperforms relative to the traditional approaches. As a result, we have introduced an ESMFold-based autoencoder. The new results show sequence-only and ESMFold-based methods perform very comparably to the best approach of one-hot encoding, achieving test accuracies on the PPI benchmark of 0.81 and 0.80 respectively, vs 0.83.
>
> * Q. The authors should be able to explain why they think the structure-aware method did not perform better than the classical methods as expected. This information will be really helpful for other works.
>
> A. The ESMfold-based structure aware method performed nearly as well as the one-hot approach and performed equivalently to the BLOSUM and VHE8 approaches despite having substantially reduced dimensionality. The underperformance of the S4pred approach could be due to the intrinsic low resolution of secondary structure as a description for proteins.

---

### Meta-Review · Area_Chair_hjpA · 2023-04-02

**Recommendation:** Invite to revise
**Confidence:** 3

**Metareview:**

This paper investigates combining pre-trained oracles with auto encoders for predicting protein structures.  In this, an auto encoder is used to generate structure-aware embeddings using the outputs of the pre-trained oracle.  The work seems original.  However, the text is not especially clear, and would greatly benefit from revising to really highlight what the inputs/outputs of each element of the model are, what embeddings correspond to and would enable etc.  The proposed method does not outperform the baselines, and while this isn't a requirement for publication here, it does raise the question of why the extra structure is justified.  Exploring the necessity and benefits of this method for downstream tasks in more detail in future revisions would greatly improve the paper.

**Summary:**

The authors investigate using auto encoders in conjunction with pre-trained oracles in a protein analysis application.  The authors suggest that good embeddings may be obtained using auto encoders.  However, the paper does not quite have the requisite clarity, underperforms existing methods, and lacks the analysis of what contributes to the performance deficit.

**Comments And Feedback To The Authors:**

I found the paper somewhat difficult to read; in particular: teasing out exactly what the method is, what the benefits of such a method would be over existing methods for downstream tasks, and what the training methodology actually corresponds to.  The auto encoder layer is also very large, and so examining smaller encoding layers would be interesting (is an encoding of roughly half the size of the original data that much of an advantage?).  Finally, comparing and validating predictions across different oracles (S4PRED vs AlphaFold) would likely improve the reach and generality of the work.

**Reason For Not Giving A Higher Recommendation:**

All three reviewers agreed that the paper is not quite ready for publication yet.

**Reason For Not Giving A Lower Recommendation:**

N/A

---

> ### Author Response · Authors · 2023-05-31
> **We have addressed the reviewers comments by rewriting the paper to make the goals more clear, reducing the size of the encoding layer, and introducing ESMFold as another oracle.**
>
> Thank you for taking your time to review our paper. We greatly appreciate your feedback and have revised our paper to address all of your comments.
>
> * Q. I found the paper somewhat difficult to read; in particular: teasing out exactly what the method is, what the benefits of such a method would be over existing methods for downstream tasks...
>
> A. Our goal is to gain comparable performance to embeddings that scale as a function of sequence length (such as one-hot encoding) with a fixed dimension representation that is on average much smaller. This should support much faster, evolutionary database scale computations for classification and regression tasks involving protein sequences (such as function or property prediction). We have rewritten the paper to make these goals more clear.
>
> * Q. ...and what the training methodology actually corresponds to.
>
> A. The training methodology is to train a sequence encoder to learn a very small, fixed dimension embedding that can either recover the sequence or aspects of protein structure that are predictable by an oracle from the recovered sequence.
>
> * Q. The auto encoder layer is also very large, and so examining smaller encoding layers would be interesting (is an encoding of roughly half the size of the original data that much of an advantage?).
>
> A. We agree with this observation and have reduced the encoding size to 100 dimensions (1-2 orders of magnitude compared to traditional embeddings) to better highlight the capability of this approach.
>
> * Q. Finally, comparing and validating predictions across different oracles (S4PRED vs AlphaFold) would likely improve the reach and generality of the work.
>
> A. We have now added new data using the protein folding network ESMFold as another oracle for comparison, and have updated the results accordingly. ESMFold was chosen over AlphaFold because ESMFold generates predictions much faster than AlphaFold, and is only marginally less accurate.

---

### Decision · Program_Chairs · 2023-04-08

Revision accepted; invite to archive

---

> ### Author Response · Authors · 2023-05-31
> **Summary of Revisions Made to the Paper**
>
> We greatly appreciate the constructive feedback from the reviewers, which has allowed us to significantly enhance the quality and scope of our manuscript. Following their suggestions, we have now substantially revised this paper.
> We introduced three new results:
> 1) S4PRED-based autoencoder: This model integrates S4PRED, an oracle for secondary structure, and we now use a significantly smaller embedding size of 100.
> 2) Pure Sequence-based autoencoder: This model focuses purely on the sequence features (embedding size of 100).
> 3) ESMFold-based autoencoder: This is a newly introduced model that was not part of our initial submission. ESMFold, an oracle for 3D structure prediction (similar to Alphafold but faster), has been incorporated into this model to enable an improved encoding that captures both sequence and 3D structural features of the proteins (embedding size of 100).
>
> These additional results allow us to further demonstrate the potential of our approach in optimizing the distillation of sequence and structural information through a hybrid approach. They also underscore the feasibility of developing task-specific structure-aware encodings to enhance the performance of encoding for specific tasks.
> Furthermore, in response to feedback, we have undertaken a comprehensive review of our text to clarify our approach and provide a more detailed exposition of our models. To improve the reproducibility of our research, we have now included more model details as well as the code that underpins our work.